# Understanding why primary care doctors leave direct patient care: a systematic review of qualitative research

Linda Long,[1] Darren Moore,[2] Sophie Robinson,[1] Anna Sansom,[3] Alex Aylward,[4] Emily Fletcher 🔵 ,[3] Jo Welsman 🔵 ,[5] Sarah Gerard Dean 🔵 ,[6] John L Campbell,[3] Rob Anderson[1]

¹ESMI (Evidence Synthesis & Modelling for Health Improvement), University of Exeter Medical School, Exeter, UK
²Graduate School of Education, University of Exeter, Exeter, UK
³Primary Care Research Group, University of Exeter Medical School, Exeter, UK
⁴Patient and Public Involvement Group, PenCLAHRC, University of Exeter Medical School, Exeter, UK
⁵Children's Health and Exercise Research Centre, University of Exeter, Exeter, UK
⁶PenCLAHRC, University of Exeter Medical School, Exeter, UK

**Correspondence to**
Dr Linda Long;
l.long@exeter.ac.uk

## ABSTRACT

**Background** UK general practitioners (GPs) are leaving direct patient care in significant numbers. We undertook a systematic review of qualitative research to identify factors affecting GPs' leaving behaviour in the workforce as part of a wider mixed methods study (ReGROUP).

**Objective** To identify factors that affect GPs' decisions to leave direct patient care.

**Methods** Qualitative interview-based studies were identified and their quality was assessed. A thematic analysis was performed and an explanatory model was constructed providing an overview of factors affecting UK GPs. Non-UK studies were considered separately.

**Results** Six UK interview-based studies and one Australian interview-based study were identified. Three central dynamics that are key to understanding UK GP leaving behaviour were identified: factors associated with low job satisfaction, high job satisfaction and those linked to the doctor–patient relationship. The importance of contextual influence on job satisfaction emerged. GPs with high job satisfaction described feeling supported by good practice relationships, while GPs with poor job satisfaction described feeling overworked and unsupported with negatively impacted doctor–patient relationships.

**Conclusions** Many GPs report that job satisfaction directly relates to the quality of the doctor–patient relationship. Combined with changing relationships with patients and interfaces with secondary care, and the gradual sense of loss of autonomy within the workplace, many GPs report a reduction in job satisfaction. Once job satisfaction has become negatively impacted, the combined pressure of increased patient demand and workload, together with other stress factors, has left many feeling unsupported and vulnerable to burn-out and ill health, and ultimately to the decision to leave general practice.

## INTRODUCTION

As described in detail previously,[1] general practice in the UK is facing a workforce 'crisis', in part due to so many general practitioners (GPs) leaving direct patient care, or reducing their hours, and many others intending to do so.[2] While this is a problem being experienced in a number of high-income countries, a report by the Commonwealth Fund in 2015 showed the problem for UK general practice

### Strengths and limitations of this study

► This systematic review was conducted and written with reference to the guidelines of the Preferred Reporting Items for Systematic Reviews and Meta-Analyses.
► Stakeholder engagement took place during the project and general practitioners on a team of co-investigators were involved in the development of the review protocol.
► Patients were involved by contributing to a patient and public involvement workshop where the explanatory model was discussed.
► Only a small number of UK studies were identified and there was limited ability to translate study findings across countries.
► Synthesis of qualitative evidence relates more or less only to National Health Service general practice in England; however, it seems likely that many of these factors are generic within primary care in the rest of the UK.

is particularly serious, with nearly 30% of GPs planning to leave general practice within 5 years.[3] In other surveys conducted between 2014 and 2016, the proportion of GPs in the UK saying they would leave general practice within 5 years varied from 29% to 42% in different regions of England.[1 4 5] The most recent (2016) UK survey, of GPs in the South West of England, showed that 70% intend to either quit, reduce their work hours or take a career break in the following 5 years.[5] At the same time GPs appear to be more stressed and more dissatisfied than ever before,[6] and more so than GPs and primary care practitioners in most other countries.[7]

We undertook a synthesis of qualitative research evidence to identify factors that affect GPs' retention in the workforce as part of a wider mixed methods study (ReGROUP) focusing on retention of experienced GPs or supporting their return to work following a career break. Through better understanding



of the factors that lead GPs—especially experienced GPs in the UK National Health Service (NHS)—to leave direct patient care, the wider ReGROUP study[8] ultimately aims to inform policies and strategies to support GPs returning to work after a career break or retain the experienced GP workforce. By identifying and analysing rich qualitative data from a variety of GP interview studies, we sought to gain a deeper understanding of why GPs are leaving UK practice and to identify and understand how factors may act individually or collectively to affect such decisions.

## Aim

This systematic review of qualitative evidence aimed to answer the following question: what are the factors in the UK and other high-income countries which affect GPs' decisions to leave direct patient care?

## METHODS

We conducted a systematic review of the qualitative literature in line with our published protocol.

### Searches

In January 2016 and March 2016, articles published in English from 1990 onwards were searched in the following databases: Medline, Medline In Process, PsycINFO, HMIC (Healthcare Management Information Consortium), Cochrane, ASSIA (Applied Social Sciences Index and Abstracts) and Web of Science (online supplementary appendix 1). We performed grey literature search including online searching, reference checking of relevant studies, and forward and backward citation searching. Further update searches were performed in May 2017 (figure 1).

### Inclusion criteria

We included qualitative or mixed methods studies which either aimed to assess factors associated with GP leaving behaviour or which are likely to have generated research data about such factors. We included studies with GPs and other primary care-based generalist doctors practising in high-income countries (online supplementary appendix 2). We sought studies which evaluated any reasons for leaving direct patient care (eg, early retirement, career

1. Family Practice/ or General Practice/
2. physicians, family/ or physicians, primary care/
3. General Practitioners/
4. Primary Health Care/
5. "primary care".tw.
6. "general practi$".tw
7. "family doctor$".tw.
8. "family physician$".tw.
9. "family medic$".tw.
10. (GP or GPs).tw.
11. or/1-10
12. (career$ adj3 (interrupt$ or chang$ or pattern$ or decision$ or leav$ or break$)).tw.
13. (retire$ adj3 (decision$ or medical$ or option$ or choice$ or pattern$ or determin$)).tw.
14. (job$ adj3 (chang$ or leav$)).tw.
15. (work$ adj3 (retention or retain$)).tw.
16. (long adj3 (sick$ or absen$ or ill$)).tw.
17. (burnout or "burn out").tw.
18. Job Satisfaction/
19. Personnel Turnover/
20. Career Choice/
21. Retirement/
22. or/12-21
23. 11 and 22
24. limit 23 to yr="1990 -Current"

**Figure 1**  Medline search strategy.

breaks, moving to hospital specialities, commissioning or public health, working part-time or never returning to work after paternal/maternal leave).

### Exclusion criteria

Sources were excluded if they were not in English language or were highly abbreviated source types (eg, conference abstracts).

### Study selection process

Titles and abstracts of search results were screened against the eligibility criteria, with an initial sample being independently screened by two authors (SR and RA) to establish consistent application of the criteria. Titles and abstracts that could not be excluded were sought as full-text articles, and the inclusion criteria applied to these (figure 2).

### Data extraction and quality appraisal

One reviewer (LL) extracted data from all published manuscripts and 50% were independently checked by a second reviewer (DM), with any discrepancies resolved through discussion. Study quality was assessed using an adapted version of the Wallace checklist[9] by one reviewer (LL) and 50% independently checked by a second reviewer (DM).

### Analysis and synthesis

Data analysis and synthesis broadly followed the principles of thematic synthesis[10] and were conducted in three stages, which overlapped to some degree: the coding of text 'line-by-line'; the organisation of these 'free codes' into related areas to construct data-driven 'descriptive themes'; and the development of theory-driven 'analytical' themes through the application of a higher level theoretical framework. Thematic analysis of textual data involved study authors' descriptions of their findings as well as primary quotations from GPs.

Of the included studies, two recent data-rich UK papers[11 12] were coded by one reviewer (LL) and the descriptive themes used to create an overall analytical framework consisting of five categories. The same two key papers were independently coded by a second reviewer (DM) and the analytical framework agreed and modified through discussion. This framework was used to code the remaining studies by one reviewer (LL), with a sample checked by a second reviewer (DM) for consistency. Data, in the form of quotations from the GPs themselves, key concepts or succinct summaries of findings, were entered into QSR's NVivo V.11 software[13] for analysis. Themes emerging from the UK studies were white-boarded and their associations considered. It was acknowledged that the identified themes could be relevant to more than one category, and this was represented in a visual 'explanatory model' (figure 3) in order to answer the review question. The model was created by one reviewer (LL) and independently checked by a second reviewer (DM), and the modifications were incorporated into the model after discussion. The model was presented and assessed

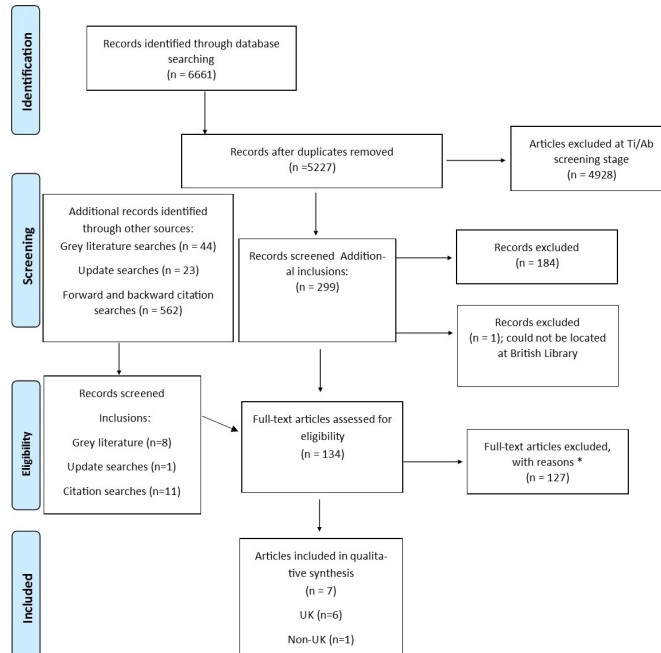

**Figure 2** PRISMA flow diagram showing the process of study selection. *Papers excluded at full-text stage are listed in online supplementary file 1online supplementary appendix 3. PRISMA, Preferred Reporting Items for Systematic Reviews and Meta-Analyses.

in terms of credibility during an involvement workshop (four patient participants) and through discussion with the wider ReGROUP project research team.

## Patient and public involvement

Patients were involved by contributing to a patient and public involvement workshop where the explanatory model was discussed (online supplementary appendix 4).

## RESULTS
### Study characteristics

Five studies (six publications) based on qualitative semi-structured interviews with practising or retired GPs were found,[11 12 14–17] all conducted in England. A further qualitative semi-structured interview study conducted in Australia was found.[18] The main characteristics of these studies are shown in table 1.

Two of the papers reporting studies from England reported findings from largely the same set of interviews,[11 12] with the later paper including a larger sample of interviewees, after intentionally recruiting more female GPs and more GPs aged 50–55 years.[12]

### Appraisal and synthesis

The analysis and synthesis presented in the following sections are based on five UK interview-based studies reported in six papers/reports.[11 12 14–17] The findings of the Australian study[18] are presented separately (online supplementary appendix 5) and discussed in relation to UK findings.

### Quality assessment

The quality of the included qualitative research studies and papers, as assessed using the 14 questions of the adapted 'Wallace tool',[9] ranged from low quality,[16] with 4 of 14 'yes' ratings on quality criteria, through to moderate quality,[14 15] with 6 of 14 'yes' ratings on quality criteria, and up to good quality,[11 12 17 18] with 9 of 14 'yes' ratings on quality criteria or better (online supplementary appendix 6).

Most studies failed to make explicit the theoretical or ideological perspective of the author (question 2). No studies provided evidence of author reflexivity (question 13). Three UK studies[14–16] and one non-UK study[18] had

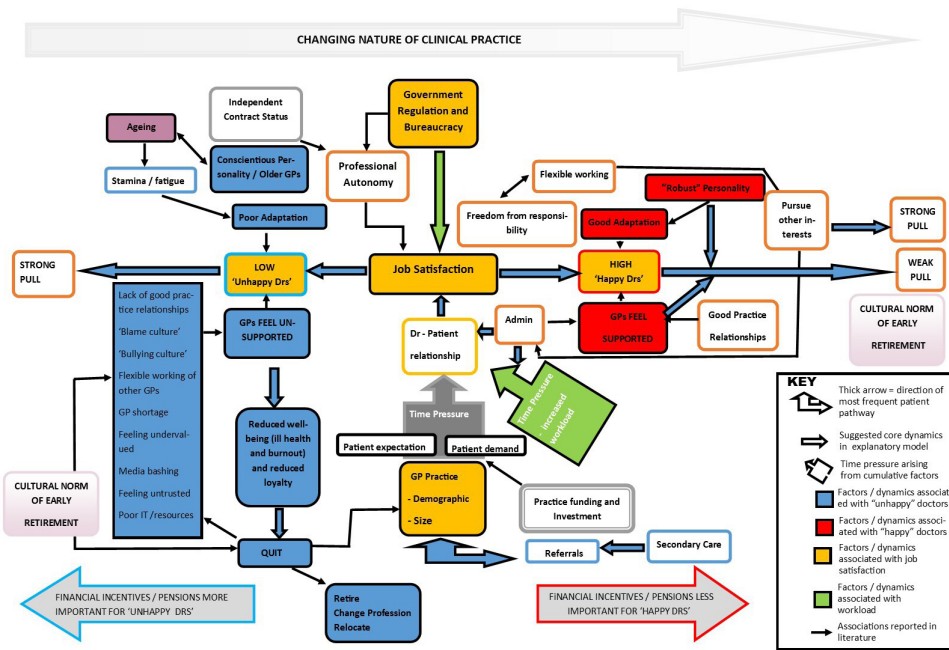

**Figure 3** Explanatory model of key factors associated with general practitioners' (GPs) leaving behaviour.

**Table 1** Characteristics of qualitative interview studies and included GPs

| Study | Year of survey(s) | Country or region | Types of GPs responding | Aim of study | GPs (n) (interview setting) | Age of GPs | % female |
|---|---|---|---|---|---|---|---|
| Doran et al[17] | NS | England | Early leavers age <50 years. | To explore the reasons why GPs leave general practice early. | 21 (by phone) | Median age band 32–54 years | 66.70 |
| Hutchins[14] | NS | England (London) | GP principals near retirement age. | Considers the reasons why many GPs are wishing to take early retirement, and measures to help retain them. | 20 (at surgery) | NS | 55 |
| Newton[15] | NS | England (Northern) | Over 45. | To describe 'Plans, reasons for, and feelings about retirement'. | 21 (at surgery or GP home, except 2 by phone) | All over 45 years | 38 |
| Sansom et al[12] | 2015 | England (South West) | Experienced GPs 50–60 years old (20 still working, 3 retired). | To investigate the reasons behind intentions to quit direct patient care among experienced GPs aged 50–60 years. | 23 (by phone)* | Age range 51–60 years | 39 |
| Campbell et al[11] | 2014–2015 | England (South West) | Experienced GPs 50–60 years old intending to retire in the next 5 years (n=14). GPs who took early retirement in the last 5 years (n=3). | To explore reasons behind GPs' intentions to quit direct patient care. | 17 (by phone)* | Age 51–60 years | 23.50 |
| | | | 15 partners, 2 locums. | | 42 (by phone) | NR | NR |
| Ipsos MORI[16] | | England | 42 GPs seriously considering leaving practice, as well as 23 GPs who had left or were in the process of returning to practice. | To identify how the experience of appraisal and re-validation might be influencing intentions to leave general practice. | 23 (by phone) | | |
| Dwan et al[18] | 2008–2009 | Australia | GPs working 6 or fewer clinical sessions per week. | To explore the nature and extent of GPs' paid and unpaid work, why some choose to work less than full time, and whether sessional work reflects a lack of commitment to the patient and the profession. | 26 (at a location determined by GP participant) | Average age: 47 years (female); 58 years (male) | 66 |

*These studies were based on largely the same sample of GP interviews. The later study (Sansom et al[12]) purposively selected more female GPs and more GPs aged 50–55 to increase the variation of age and sex across the sample.
GP, general practitioner; NS, not stated.

**Table 2** Analytical framework showing identified categories and themes around general practitioners' decisions to leave direct patient care

| Undoable/unmanageable | Morale | Impact of organisational changes |
|---|---|---|
| Workload | Identity/perceived value | Referrals |
| Pressures | Professional culture | Targets and assessments |
| Fear of making mistakes | Lack of support | Doctor–patient relationship |
| Training and resources | Government/political | Changing role |
| Patient demands | Wider community | Autonomy and control |
| Practice demands | Negative 'media-bashing' | Re-accreditation |
| | Job satisfaction | |
| | Well-being | |
| | Work–life balance | |
| **Projected future** | **Multiple options and strategies** | |
| Viability (of early retirement) | Flexible working | |
| Ageing | Continue and cope | |
| Investment and commitment | Alternative roles | |

further limitations in relation to two to four other quality criteria.

All of the themes in the synthesis were informed by at least two studies, and there was at least one good quality study informing every theme (online supplementary appendix 7). The low-quality to moderate-quality UK studies alone did not determine any of the themes, but did provide support for them.

### Categories and themes

The synthesis consisted of a series of linked themes affecting whether GPs leave direct patient care or reduce their time commitment to patient care, each of which belongs to one of the five categories summarised in the analytical framework in table 2 and the full details are given in online supplementary appendix 7.

These categories from the qualitative synthesis were, first, GPs experiencing working as a GP as 'undoable and unmanageable'. Many GPs are experiencing working as a GP as undoable and unmanageable due, among other reasons, to high/increasing administrative workload and high/increasing patient demand (both in the number of patients, and their complexity and higher expectations), together with a perceived lack of training and resources to cope with these pressures.

The second category, 'low morale', was seen to be associated with reductions in the perceived value of GP work (with loss of identity) and changed professional culture (more target-driven and standards-driven rewards system; multidisciplinary team-based working (yet for some also lone working/isolating culture); a more aggressive top-down managerial culture within the NHS; and more widespread norms and expectations for early retirement). Low morale was seen as associated with a lack of support from both the government and political parties, and the negative portrayals of GPs by news media. Morale was also seen to be closely linked with job satisfaction (or

dissatisfaction), neglect of personal well-being/health and feelings about work–life balance.

The third category was the 'impact of organisational changes'. The perceived key factors under this theme were changes in referrals—both restricted opportunities to refer to secondary care, and higher numbers of (and more complex) referrals from secondary care—as well as a greater focus on targets and assessments, and fears about re-accreditation (including evidence that some GPs might retire early in order to avoid re-accreditation). Some of the organisational changes were considered to have imposed increased clinical and non-clinical responsibilities and work on GPs. Together, such changes were believed to have undermined some of the basic tenets and traditional expectations of being a GP, such as the doctor–patient relationship and having autonomy and control over one's clinical work.

The fourth category was how GPs projected their future, which related to ageing, the financial viability of reducing hours or retiring early, and to what extent GPs were personally committed and financially invested in their practices. These included problems linked to whether younger GPs wanted to take on the responsibility of becoming practice partners, and also possible tensions between older and younger GP partners (in the way practices are run, in major investment/refurbishment decisions, or in relation to planning for partners retiring and needing new partners to buy out their share of a practice).

Finally, the fifth category was called 'multiple options and strategies' and referred to the various ways in which GPs either continue and cope or—perhaps if less committed or less resilient, or if they can simply afford to financially—decide to leave or go part-time. This theme also highlighted the major importance of flexible working, that is, working reduced hours (eg, by becoming a locum) as a method of coping and regaining work–life balance



and job satisfaction. For others, the adoption of alternative work roles outside general practice, often part-time, allowed use and learning of other skills—either as relief and variety from working as a GP, or for some as a potential alternative career. The kinds of alternative roles and options GP interviewees mentioned included becoming complementary therapists, CCG (Clinical Commissioning Group) leads, advisory committee members, or working for pharmaceutical consultancies or teaching in medical schools. Like part-time working, for some these might be clear routes for quitting general practice, but for others such variety of roles and opportunities for job satisfaction may keep them in general practice.

### Explanatory model and narrative summary of key factors influencing UK GPs

Themes were used to construct an explanatory model (figure 3). This model makes it possible to 'go beyond' the findings of the primary studies and generate additional concepts, understandings and hypotheses relating to factors influencing GPs' decisions to quit general practice. 'Real world' applicability was confirmed following feedback on the model from patients and project stakeholders during face-to-face discussions in a stakeholder meeting.

Above the explanatory model (in grey), the changing nature of general practice over time is presented separately, providing a contextual lens from which to view the main model. The career path and expectations of UK GPs have changed considerably over the last 40 years. Today's GP is expected to be a member of a wider multidisciplinary team commissioned to deliver national standards of care and has a role barely recognisable to the one many experienced GPs practising in the 1990s remember, where GP partners tended to stay in one practice for most of their career and there was less regulation and a high expectation of autonomy. In the contemporary career model, GPs said they are expected to give up autonomy in many areas of their job and are expected to accommodate increasing government regulation and bureaucracy, which increases stress related to workload, particularly 'paperwork'/record-keeping.

Factors associated with job satisfaction (shaded orange in figure 3) are listed along with factors associated with high job satisfaction on the right (shaded red) and factors associated with low job satisfaction on the left (shaded blue). Job satisfaction appears pivotal to whether a GP will successfully adapt and remain in practice, or will become overwhelmed by external influences and pressures and leave the profession. GPs said job satisfaction directly relates to the quality of the doctor–patient relationship, with more time available for GPs to spend with their patients being associated with better job satisfaction. GPs with high job satisfaction describe feeling supported by good practice relationships, while GPs with low job satisfaction describe low morale and feeling unsupported.

Some GPs experiencing low job satisfaction report a lack of good practice relationships and describe working in a 'blame culture' where they fear litigation.[17] Others describe a 'bullying culture', and feel undervalued and mistrusted by patients and government, in addition to being inadequately trained in information technology (IT), under-resourced and poorly portrayed in the media.[17] Older GPs or GPs with a more conscientious personality may find it more difficult to adapt, and some GPs describe physical symptoms of fatigue and loss of stamina, for example, women experiencing sleeplessness due to menopause.[11] GPs with low job satisfaction appeared more likely to experience reduced feelings of well-being and experience ill health and burn-out.[11] They were also less likely to experience feelings of loyalty to the NHS and more likely to quit (retire, change profession or relocate), exacerbated by a cultural norm of early retirement in the profession.[11] Financial incentives and pension arrangements appeared to be more important to GPs with low job satisfaction, and for some GPs financial incentives (intended to help retain GPs) may cause them to retire earlier rather than stay in practice longer.[15]

GP shortages (through poor recruitment and retention) and patient demand are creating pressure on full-time GPs, leading some to consider retiring. Patient demands may be higher in areas of higher deprivation and with populations with multiple health and social problems.[11] The impact of GP shortages is most keenly felt in smaller practices, with some GPs feeling trapped between continuing to work full time under extreme pressure or to retire completely as they fear working part-time would shift the burden of responsibility onto colleagues.[11] The explanatory model shows how this situation is compounded by pressures from increased workload (figure 3, shaded green), particularly from increased administration as well as from secondary care.[12] Increased complexity in referral pathways, for example, hospitals providing increasingly specialised services (ie, shifting more care to primary care) and delays in communication, contributes to GPs experiencing a depersonalised, fragmented healthcare system.[17] Feelings of uncertainty over the future of general practice are prevalent, with GPs less likely to invest in building and make long-term commitments.[11] Younger GPs may be more reluctant to take on partnerships because of the added responsibilities and risks involved. For some, poor relationships between older and younger doctors and/or opposing views about how a practice should be run result in older GPs feeling unsupported, less loyal to the NHS and more likely to leave.[12]

In summary, UK GPs with poor job satisfaction report feeling overworked and unsupported. Combined with changing relationships with patients and interfaces with secondary care, and the gradual sense of loss of control over large parts of the job, many GPs report a reduction in job satisfaction. Lack of time with patients is perceived to compromise the ability to practise patient-centred care and undermines GPs' professional autonomy and values, resulting in further diminished job satisfaction. Once job satisfaction has become negatively impacted,

the combined pressure of increased patient demand and workload, together with other stress factors such as poor IT resources, negative media portrayal, poor practice relationships, and a 'bullying' or 'blame' culture, has left many feeling unsupported and vulnerable to burn-out and ill health, and ultimately to the decision to leave general practice.

## DISCUSSION

The thematic analysis of four qualitative interview studies with UK GPs, two from 2015 and 2016 and two older ones from 2004 and 2005, yielded five overarching types of factors related to GPs leaving or intending to leave direct patient care or reduce their hours, together with more specific sub-themes underlying or linked to these five factors. These themes were categorised into a framework, and the relationships between identified factors were summarised in a visual explanatory model that was developed from them (figure 3). All of these qualitative studies were judged to be of reasonable to good quality.

Overall, the rather negative picture portrayed by the four qualitative interview studies was that UK GPs with poor job satisfaction are also those who feel overworked and unsupported. Many feel part of an over-bureaucratised system and describe being at the front end of a service unable to deliver what it promises. Combined with changing relationships with patients and changing interfaces with secondary care, and the gradual sense of loss of control over large parts of the job, many GPs report a reduction in job satisfaction over time. Lack of time with patients is perceived to compromise the ability to practise patient-centred care and continuity of care and with it the GPs professional autonomy and values, resulting in diminished job satisfaction. Once job satisfaction has become negatively impacted, the combined pressure of increased patient demand and workload, together with other stress factors such as poor IT resources, negative media portrayal, poor practice relationships and a perceived 'bullying' or 'blame' culture, has left many feeling unsupported and vulnerable to burn-out and ill health. Ultimately, for some this leads to their decision to leave general practice altogether or to substantially reduce their clinical hours.

Our explanatory model (figure 3) highlights the pivotal role of administrative support in enabling GP flexible working. Both Hutchins[14] and Doran et al[17] support this finding, suggesting that additional administrative assistance could enable more time to see patients. Given that our synthesis indicates that having sufficient time to see patients is a significant driver of GP job satisfaction, and that job satisfaction is strongly associated with GP retention, increased administrative support may offer a simple solution to the problem of GP retention in the UK. However, it is unlikely that this step alone will solve the problem. Our explanatory model also highlights the complexity of the problem and suggests solutions for retention will not be simple. This is supported by Ipsos MORI,[16] which states there can be no 'silver bullet' approach to the complex multi-factorial issues underlying current disaffection among UK GPs.

### Strengths and weaknesses
#### Strengths
This systematic review was conducted and written with reference to the guidelines of the Preferred Reporting Items for Systematic Reviews and Meta-Analyses. Potential for transferability of findings to UK practices is based on stakeholder engagement during the project. Relevant stakeholders were involved in the review; several GPs on a team of co-investigators were involved in the development of the review protocol.

The author team consists mainly of academic health researchers employed by the University of Exeter, with one of the authors (AA) being a patient representative. One of the academic health researchers (JLC) has previously worked in the NHS as a GP, while another (SGD) has previously worked in the NHS as a physiotherapist.

#### Limitations
Limitations include identification of a small number of UK studies. Although a single non-UK study was identified (not reported here), we were not able to translate study findings across countries. In addition, the synthesis of qualitative evidence presented here relates more or less only to NHS general practice in England. However, it seems likely that many of these factors are generic within primary care in the rest of the UK. We acknowledge that there are limitations from conducting a secondary analysis without coding original transcripts from these studies. Also, of the good quality studies that informed the themes in the synthesis, none explicitly provided a theoretical or ideological perspective of the author (or funder) and none of the authors was reflexive, and these limitations may influence individual study research findings and hence the themes identified in this synthesis.

### CONCLUSIONS
While recognising the complexity of the current situation, and acknowledging there is unlikely to be a 'silver bullet' solution, the synthesis shows an association between flexible working and improved job satisfaction, potentially delaying retirement. GPs' views suggest that the stress associated with seeing more patients, including more complex patients, but with the same traditional constraints on appointment times, needs to be addressed. Solutions involving alleviating non-clinical administrative burden, for example, through additional staff resources resulting in more patient-centred care, may be motivating to many GPs.

**Acknowledgements** We are very grateful to Simon Briscoe and Chris Cooper for their generous support at the earlier stages of planning this review and for supporting the project's information specialist (SR). We thank the rest of the

ReGROUP project coinvestigators and researchers for their support and useful comments in progress meetings. We would also like to thank all the PPI group members who attended the mid-review PPI workshop and provided valuable feedback and comments on our emerging review findings and explanatory model. SGD is also supported by the National Institute for Health Research (NIHR) Collaboration for Leadership in Applied Health.

**Contributors** LL, DM, SR, RA, AS, AA, EF, JW, SGD and JLC made substantial contribution to the conception and/or design of the work. LL, SR, AS, AA and JW contributed to the acquisition, analysis and interpretation of data for the work. LL, DM, SR, RA, AS, AA, EF, JW, SGD and JLC provided input to drafting the work and/ or revising it critically and gave final approval of the version to be published. All are accountable for all aspects of the work in ensuring that questions related to the accuracy or integrity of any part of the work are appropriately investigated and resolved.

**Funding** The project was funded by the National Institute for Health Research HS&DR programme (project 253 14/196/02). The views and opinions expressed are those of the authors and do not necessarily reflect those of the HS&DR programme, the NIHR, the NHS or the Department of Health.

**Competing interests** None, except that two of the included studies were conducted by two of the coauthors of this systematic review (JLC and AS) and the principal investigator of the wider ReGROUP study of which this systematic review is a part (JLC). Neither AS or JLC had any involvement in the detailed data extraction or quality assessment of their studies or any of the other studies. Also, AA has received personal fees from the Northern, Eastern and Western Devon CCG, Devon Local Medical Committee, British Medical Association, University of Exeter, CLAHRC South West Peninsula, and NHS England Medical Directorate (South), outside of this work.

**Patient consent for publication** Not required.

**Provenance and peer review** Not commissioned; externally peer reviewed.

**Data availability statement** All data relevant to the study are included in the article or uploaded as supplementary information. No additional data available.

**ORCID iDs**
Emily Fletcher http://orcid.org/0000-0003-1319-3051
Jo Welsman http://orcid.org/0000-0002-8877-3926
Sarah Gerard Dean http://orcid.org/0000-0002-3682-5149

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
