## [Reviewer comments · BMJ Open]

ARTICLE DETAILS

TITLE (PROVISIONAL)	Understanding why primary care doctors leave direct patient care - A systematic review of qualitative research
AUTHORS	Long, Linda; Moore, Darren; Robinson, Sophie; Sansom, Anna; Aylward, Alex; Fletcher, Emily; Welsman, Jo; Dean, Sarah; Campbell, John; Anderson, Rob

VERSION 1 – REVIEW

REVIEWER	Khatijah Abdullah University of Malaya
REVIEW RETURNED	14-Mar-2019

GENERAL COMMENTS	Thank you for an interesting paper However although the abstract was clear the write up of the methodology and findings were lengthy and confusing The proposed model was not well explained and confusing.
---

REVIEWER	JENS SØNDERGAARD Research Unit for General Practice, Department of Public Health, University of Southern Denmark
REVIEW RETURNED	09-Apr-2019

GENERAL COMMENTS	The manuscript is well written and address an important topic. I suggest some minor clarifications: 1. The background and prejudices of the authors should be explained 2. Can the authors say something about the importance of each barrier and facilitator? 3. The authors may consider to be more direct when they discuss implications of their findings
--

REVIEWER	Dr. Emily Gard Marshall Dalhousie University Department of Family Medicine Canada
REVIEW RETURNED	03-Sep-2019

GENERAL COMMENTS	I commend the authors on conducting this synthesis of qualitative research; a method that is both valuable and under-represented in health services research. The topic of why primary care doctors leave direct patient care is also very prescient due to concerns about rising challenges with access to primary care and family physician's burnout in many countries. Overall, this manuscript is thoughtful and well written with abundant supporting appendices. This was clearly a great deal of work to review so many articles and conduct this new analysis. The work is strengthened by a clear and systematic
--

appraisal and synthesis method. I commend the use of both provider and patient stakeholders in a confirmatory capacity for the explanatory model (perhaps this could be referenced as a face validity method?)

A few recommendations may improve the precision, clarity and helpfulness of this manuscript:

1. Recommended: The abstract objectives call the work a “synthesis of qualitative research” while the title and methods refer to a systematic review. Consistency would help clarify.

2. Required: Under article summary, bullet 2 “basis for transferability...” may help to specify transferability within what context. UK?

3. Required: There is a one paragraph introduction, followed by aims, then methods. It would be helpful, particularly for an international audience, to have a background section that clarifies the basis for the assumption that the “British GP workforce is said to be “crisis” and the general state of knowledge on the topic. Helpful to include how the UK situation related to what is happening in other jurisdictions. I would anticipate that much of that literature would also include increasing rates expected retirements due to demographics of physician population? By acknowledging these pieces in the background, it sets up the need for this qualitative review analysis to answer questions about the less-expected reductions in family practice. Pg. 4

4. Recommended: the original literature review was completed in spring of 2016 with an update in May 2017. That leaves a gap in time for any newer relevant studies to be included. Though I recognize that it would be cumbersome to look for new literature and incorporate new findings into the analysis. However, it would strengthen the timeliness and relevance if you chose to do so.

5. Required: Under methods, please clarify/choose between extracting and analyzing “studies” vs. “published manuscripts”. It was unclear if this team’s analysis was of the published papers only or they coded the original qualitative data of those studies.

6. Required: in the Categories and Themes section, it would benefit from linking the coding back to the articles. There is a statement in the methods that each theme had at least 2 high quality sources. These could either be referenced within the section or a table could be added. Pg. 11

7. Recommended: For the explanatory model, you may want to reference the method of confirming feedback that was provided by stakeholder as face validity. Pg. 12, para 1

8. Required: Figure 3 Explanatory Model of key factors associated with GP leaving behaviour. First, I applaud the development of this model and agree that a model representing the findings and their inter-relationships may hold great insights in how to understand and address the core issue under study. There are some challenges with the current figure that I believe will hinder its effectiveness, but could be addressed with some modifications. First, the figure is very difficult to read, even when I enlarged it on my large screen. The figure could use the assistance of a graphic designer. The text is often not centered or consistent in placement. It is also overly complex. Perhaps seeing if there are sections/lists that could be combined/summarized and reduce the busy-ness. Imagine putting this table up on a screen and presenting to an audience of family physicians and policy-makers. I fear the complexity would turn the audience away and reduce the chance that they would hear and understand the key messages. The concept is great. My advice is to refine. This would also help with readability.

	9. Again, page 14 would benefit from references to the papers or a table. 10. Required: Strengths and weaknesses. Please add the limitations of some of the qualitative methods used in the papers cited as discussed in page 10 of your quality assessment. Discuss these limitations and how they, and a secondary analysis without coding original transcripts from those studies, may have implications for these findings. P. 15 Thank you for this work. I look forward to the opportunity to review a revised version.
--	--

VERSION 1 – AUTHOR RESPONSE

We provide the following responses:-

- While we felt it was important to retain the detail required for good reporting of methods and results, we have now shortened and simplified the methods and findings sections. We have now simplified the proposed model to aid clarification and understanding of the explanatory text.

- Background and potential prejudices of authors are now addressed in the Discussion section under 'limitations'. Associated prejudices are detailed again in the 'Competing interests' section of the submission

- The influence of facilitators on retention, as identified in Fig 3, are described on p.12 (last paragraph) and while the influence of barriers, as identified in Fig 3, are described on p.14.

- In the discussion, we have now directly highlighted the potential for additional administrative assistance to enable more time to see patients which may lead to increased job satisfaction and retention.

- 'synthesis of qualitative research' has now been replaced by 'systematic review of qualitative research' in the abstract.

- We have now clarified that we are talking about potential for transferability of findings to UK practices.

- We have provided a completely expanded and revised opening introductory paragraph, citing 6 new references.

- Thanks for this useful suggestion. To address them, a completely expanded and revised opening introductory paragraph is now included, citing 6 new references.

- While we recognise the potential value of running additional searches to ensure any more recent literature is included, if we found any we would not unfortunately have the time to incorporate it in our synthesis. In addition, our thematic analysis yielded a repetition of themes between studies. Indicating that we had achieved saturation with respect to themes, suggesting that further studies would be less likely to add further themes and extend our understanding.

- Data was extracted from all included published manuscripts. Line one of 'Data extraction and quality appraisal' has been amended in track changes accordingly.

- Face validity was gained through face-to-face discussion at a stakeholder meeting. This has been added in track changes to 'explanatory model and narrative summary of key factors influencing UK

GPs'

- Figure 3 has now been simplified.

- We acknowledge that there are limitations from conducting a secondary analysis without coding original transcripts from these studies. Also, of the good quality studies that informed the themes in the synthesis, none explicitly provided a theoretical or ideological perspective of the author (or funder) and none of the authors were reflexive which may influence research findings. These limitations have been added in track changes in the discussion under 'Limitations'.

VERSION 2 – REVIEW

REVIEWER	JENS SØNDERGAARD Research Unit for General Practice, Department of Public Health, University of Southern Denmark
REVIEW RETURNED	30-Jan-2020

GENERAL COMMENTS	The authors have addressed the editors' comments appropriately.
---

REVIEWER	Emily Gard Marshall Dalhousie University Canada
REVIEW RETURNED	24-Jan-2020

GENERAL COMMENTS	I thank the authors for a conscientious revision of the manuscript. I am satisfied that they have appropriately responded to the reviewers recommendations. I recommend this manuscript for publication.
--